# Recovery of 2R.2S$^k$ Triticale-*Aegilops kotschyi* Robertsonian Chromosome Translocations

**Waldemar Ulaszewski [1]**, **Jolanta Belter [1]**, **Halina Wiśniewska [1]**, **Joanna Szymczak [2]**, **Roksana Skowrońska [2]**, **Dylan Phillips [3]** and **Michał T. Kwiatek [1,*]**

[1] Department of Genomics, Institute of Plant Genetics of the Polish Academy of Sciences, Strzeszyńska 34, 60-479 Poznań, Poland; wula@igr.poznan.pl (W.U.); jbel@igr.poznan.pl (J.B.); hwis@igr.poznan.pl (H.W.)

[2] Department of Genetics and Plant Breeding, Poznań University of Life Sciences, Dojazd 11, 60-632 Poznań, Poland; asia.szymczak28@wp.pl (J.S.); roksana.skowronska@up.poznan.pl (R.S.)

[3] Institute of Biological, Environmental and Rural Sciences, Aberystwyth University, Aberystwyth, Ceredigion, Wales SY23 3DA, UK; dwp@aber.ac.uk

[*] Correspondence: michal.kwiatek@up.poznan.pl; Tel.: +48-61-848-7760

**Abstract:** Robertsonian translocations (RobTs) in the progeny of triticale (×*Triticosecale* Wittmack) plants with monosomic substitution of *Aegilops kotschyi* chromosome 2S$^k$ (2R) were investigated by fluorescence in-situ hybridization. Chromosome 2S$^k$ of *Ae. kotschyi* is reported to possess many valuable loci, such as *Lr54 + Yr37* leaf and stripe (yellow) rust resistance genes. We used a standard procedure to produce RobTs, which consisted of self-pollination of monosomic triticale plants, carrying 2R and 2S$^k$ chromosomes in monosomic condition. This approach did not result in RobTs. Simultaneously, we succeeded in producing 11 plants carrying 2R.2S$^k$ compensatory RobTs using an alternative approach that utilized ditelosomic lines of triticale carrying 2RS (short arm) and 2RL (long arm) telosomic chromosomes. Identification of molecular markers linked to *Lr54 + Yr37* genes in the translocation plants confirmed that these resources can be exploited in current triticale breeding programmes.

**Keywords:** *Aegilops*; centric breaks; chromosome fusion; Robertsonian translocations; telosomic chromosomes; triticale

## 1. Introduction

Hexaploid triticale (×*Triticosecale* Wittmack, $2n = 6x = 42$, AABBRR) is one of the few artificial crops cultivated at a large scale. Triticale combines the quality traits of wheat (*Triticum aestivum* L.) and the adaptation abilities of rye (*Secale cereale* L). Over the last three decades the global harvested area of triticale has constantly increased (2,101,405 ha in 1996; 3,662,363 ha in 2006, and 4,157,018 ha in 2016) [1], and the range of uses has also grown. Forage production is the principal end use for this crop, but there are new niches proposed, such as: biofuel production [2,3], baking [4], brewing [5] and food production [6]. As a recent man-made crop, triticale suffers narrow genetic variability for breeders to select upon.

In triticale there has been a breakdown of resistance against leaf rust (caused by *Puccinia triticiana* Eriks.) and stripe (yellow) rust (caused by *P. striiformis* f. sp. *tritici* Westend.). As the harvested area of triticale has increased, new pathotypes of *Puccinia* have evolved, moving from wheat and rye into triticale [7]. Both pathogens can reduce the grain yield by 40% [8,9]. There is an urgent need to improve the genepool of triticale, and introduce genetic resistance to *Puccinia* infections. There are approximately eighty leaf rust resistance genes identified in Triticeae, and nearly the same amount of stripe rust resistance genes have also been identified. Some of these genes have already been transferred

from wild relatives into the wheat genetic background [10]. Recently, several attempts were made to transfer rust resistance genes from *Aegilops*, *Agropyron* and *Triticum* species into triticale [11–15].

*Aegilops* species are closely related to wheat (and triticale, *per se*) and carry a number of valuable traits, which have been effectively incorporated into wheat by developing wheat–*Aegilops* hybrids and deriving addition, substitution and translocation lines [16]. *Aegilops kotschyi* Boiss. ($2n = 4x = 28$ chromosomes, U- and S-genomes) is a wild tetraploid goatgrass native to Northern Africa, the Mid-East, and Western Asia. *Ae. kotschyi* germplasm is exploited in wheat breeding [17] as a source of high grain protein, iron and zinc [18]. Moreover, Antonov and Marais [19] observed leaf rust resistance that was effective against the infection of *Puccinia triticina* in *Ae. kotschyi*. Marais et al. [20] identified the *Lr54* and *Yr37* leaf rust and stripe rust resistance genes, and developed aT2DS.2S$^k$L wheat-*Ae. kotschyi* translocation line. The first *Lr54 + Yr37* marker was developed by Heyns et al. [21]. Moreover, translocation gene sequences were cloned and specific SSR markers were developed [22].

Homoeologous recombination based engineering is the most common way for efficiently utilizing the wild relative gene pool for crop improvement [23]. The generation of translocation lines is the most promising pathway for the exploitation of alien germplasm in crop breeding [23]. In distant hybrids, unpaired chromosomes are present as univalents during meiosis. Monosomic chromosomes are prone to centric breaks at anaphase I of meiosis, which misdivide and the broken ends fuse during the interkinesis of meiosis II [24–26]. Fusion of the misdivided products may result in the formation of a Robertsonian translocation (RobT) [27].

Several steps are required to generate RobTs (Figure 1a), with self-pollination of double-monosomic plants being the most common method used in the induction of RobTs [26]. Wheat breeders can use a large collection of aneuploid stocks for the induction of wheat-wheat and wheat-alien RobTs. This can be performed in a directed manner by producing the appropriate wheat or alien monosomic lines [28–31]. In wheat breeding the most common RobTs are the T1BL.1RS and T1AL.1RS translocations [25]. Lukaszewski and Curtis developed 1RS.1DL chromosome translocation in triticale "Rhino" and transferred this to cv. "Presto" by backcrossing for several generations [32]. The 1RS.1DL translocation was later used for the induction of multi-breakpoint translocation lines [29,33,34]. A key issue for the centric breakage-fusion mechanism is the frequency at which the centric breakage events involving the univalent occurs. Friebe et al. [26] reported that this can range from 1% to 11%. Another issue is the resultant telocentric chromosomes have a tendency of rejoin as RobTs. It was reported that the frequency of wheat–alien RobTs recovered ranges from 4% to 20%, depending on the genetic background, the chromosomes involved, and environmental conditions used [26,30,31].

To increase the rate of RobT formation, it is therefore necessary to reduce the random factors connected with appearance of centric breaks. In this study, we postulated that the use of telosomic plants for cross-hybridization with plants carrying an alien-substitution will overcome the random process of centric break formation in the univalent (Figure 1b). This approach was used to transfer the *Lr54 + Yr37* resistance genes into triticale cv. "Sekundo" through a T2RS.2S$^k$L RobT. It was hypothesized that the presence of the 2R donor chromosomes in a telosomic condition would increase the frequency of RobTs recovery (Figure 1b). To test this we used a monosomic substitution triticale line carrying a single 2S$^k$ chromosome, instead of a 2R triticale chromosome (40 + M2R + M2S$^k$), and crossed this with a triticale line carrying a single copy of 2RS (short arm of 2R chromosome) and 2RL (long arm of 2R chromosome) telosomic chromosomes (40T + D2RS + D2RL). As a control experiment, the classical approach, based on the self-pollination of double monosomic triticale-*Ae. kotschyi* plants (40T + M2R + M2S$^k$), was also undertaken (Figure 1a). The goal was to compare the frequencies of chromosomal breaks and recovery of RobTs that arose throughout the two independent approaches tested (Figure 1a,b).

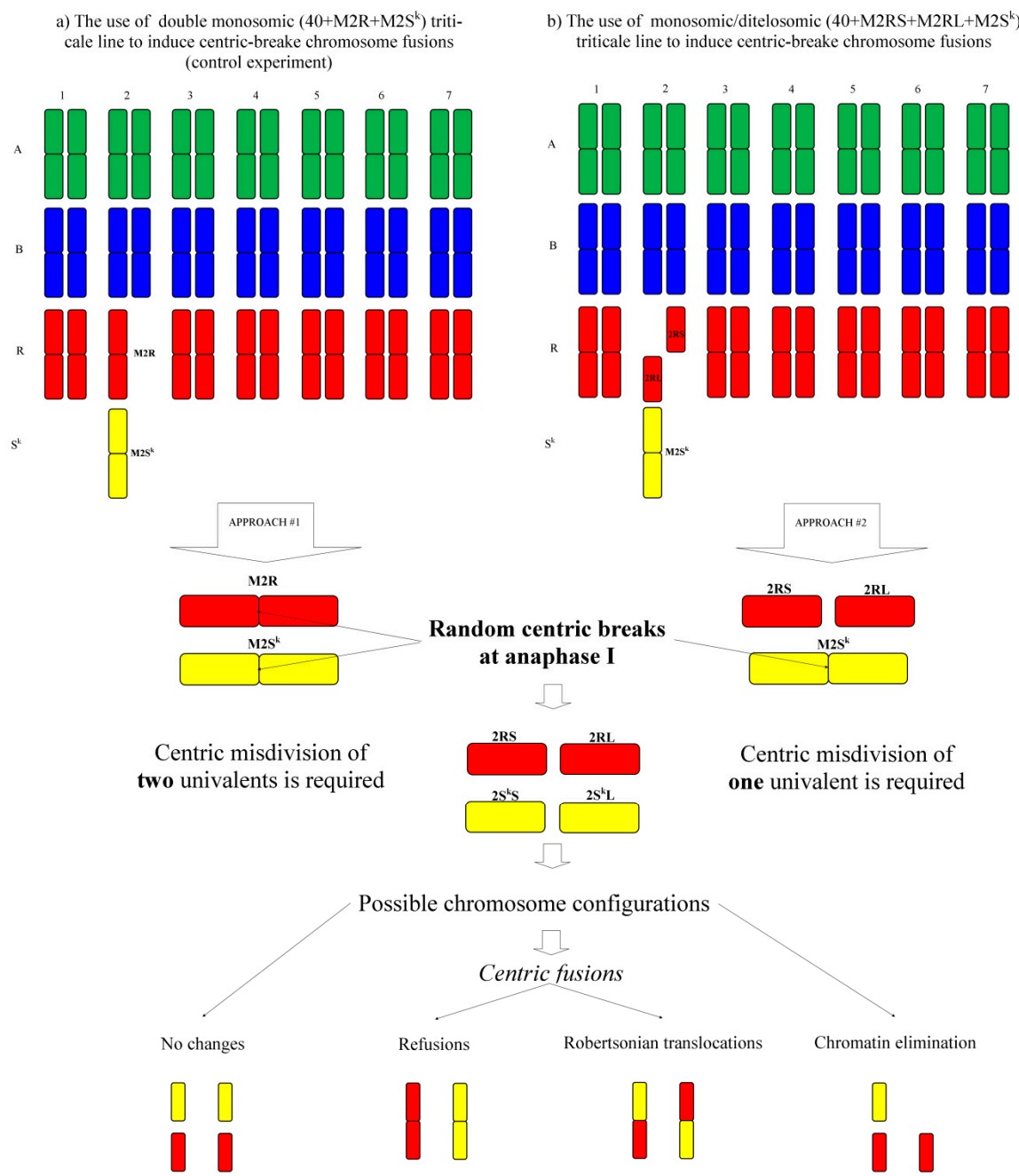

**Figure 1.** Ideogram of the two contrasting approaches for the recovery of Robertsonian translocations.

## 2. Materials and Methods

### 2.1. Plant Material

Winter triticale cv. "Sekundo" was obtained from the Danko Hodowla Roślin Sp. z o.o. (Choryń, Poland) plant breeding company. Accession Ak3 of *Aegilops kotschyi* was received by Prof. B. Wojciechowska (Institute of Plant Genetics of the Polish Academy of Sciences, Poznań, Poland; IPG PAS) from a collection of Professor M. Feldman (Weizmann Institute of Science, Rehovot, Israel) and assembled in the genebank of the IPG PAS. All plants were grown and crossed in the greenhouse of the IPG PAS. In the initial study, monosomic addition triticale line (MA2S$^k$; $2n = 43$) that carried chromosome 2S$^k$ derived from triticale-*Aegilops kotschyi* hybrids was used. These plants were crossed with triticale nullisomic plants for chromosomes 2R (N2R, $2n = 40$); obtained

by Kwiatek et al. [35]. Double monosomic (M2R and M2S$^k$) F$_1$ plants (2*n* = 42) were identified using GISH, according to Kwiatek et al. [35]. Ditelosomic (Dt2RS and Dt2RL) plants of triticale were generated by Kwiatek et al. [35] by harnessing chromosome fragmentation induced by gametocidal action, in combination with the production of double haploids. Spikes emasculated between April and June of 2018 were used for cross-pollination. The anthers of donor plants were collected, and pollen was transferred onto stigmas by chafing.

## 2.2. Chromosome Preparation

Accumulation of cells at mitotic metaphase and fixation was carried out according to Kwiatek et al. [35]. The root meristemes were digested at 37 °C for 2 h and 40 min in an enzymes solution containing 0.2% (*v/v*) Cellulase Onozuka R-10 and Calbiochem cytohelicase (1:1 ratio) and 20% pectinase (Sigma), in 10 mM citrate buffer (pH 4.6). Chromosome preparations were made according Heckmann et al. [36]. Digested root tips were placed on slides with a drop of ice-cold 60% acetic acid. The mixture was spread on a slide using metal needle for 2 min on a heating table (Medax) set at 48 °C. The slides were washed with 200 μL of ice-cold ethanol-acetic acid (3:1, *v/v*) and placed in 60% acetic acid for 10 min, washed in 96% ethanol, and air dried.

Meiotic chromosome spreads from pollen mother cells were prepared from flower buds fixed with ethanol-acetic acid (3:1, *v/v*) according to Zwierzykowski et al. [37]. Anthers were transferred to a slide with a drop of ice-cold 60% acetic acid and dispersed with a metal needle. The slide was placed on a hot plate (45 °C) for 2 min, during which the drop was spread using a needle. The slide was removed from the hot plate, a drop of 60% acetic acid added and covered with a cover slip. The slide was frozen in liquid nitrogen, and cover slip was removed with a razor blade.

## 2.3. Probe Preparation and Fluorescence in Situ Hybridization

Total genomic DNA was isolated using GeneMATRIX Plant & Fungi DNA Purification Kit (EURx, Gdansk, Poland). DNA of *Aegilops sharonensis* Eig. (a progenitor of the S-genome of *Ae. kotschyi*; PI 551020, U.S. National Plant Germplasm System, Aberdeen, ID, United States of America) was labeled by nick translation with Atto-488 dye (Atto-488NT kit; Jena Bioscience, Jena, Germany) to investigate *Aegilops* chromosomes behavior during meiosis in the hybrid. Blocking DNA from triticale (Sekundo, Lamberto and Bogo) was sheared by boiling for 30-45 min and used at a ratio of 1:50 (probe:block). Genomic in-situ hybridization (GISH) or multicolor GISH was carried out according to previously published protocols [35]. Mitotic chromosomes were identified using fluorescence in situ hybridization (FISH) with the repetitive sequences from pTa-86, pTa-535, pTa-465, pTa-k-566 and pTa713 clones characterized by Komuro et al. [38]. They were amplified from genomic DNA of wheat (Chinese Spring) according to Kwiatek et al. [39], and labelled with Atto-488, Atto-550, and Atto-647 dyes using a nick translation kit (Jena Bioscience). FISH was performed according to Kwiatek et al. [35]. Slides were examined with the Olympus BX 61 automatic epifluorescence microscope equipped with Olympus XM10 CCD camera. Image processing was carried out using Olympus Cell-F (version 3.1; Olympus Soft Imaging Solutions GmbH: Münster, Düsseldorf, Germany) imaging software and PaintShop Pro X5 software (version 15.0.0.183; Corel Corporation, Ottawa, ON, Canada). Chromosomes of *Aegilops* and triticale were identified by comparing the signal patterns of the probes [39].

## 2.4. Lr54 + Lr37 SSR Marker Screening

Genomic DNA of parental forms and offspring plants were isolated using Plant DNA Purification Kit (EurX Ltd., Gdańsk, Poland). All primers (Table 1) were manufactured by Sigma-Aldrich (Merck).

PCR reactions were performed in a LabCycler thermal cycler (SensoQuest Biomedizinische Elektronik, Goettingen, Germany). The 20 μL PCR reaction consisted of 150 nM each primer, 0.2 mM of each nucleotide, 1.5 mM MgCl$_2$, 0.2 units of Taq-DNA hot-start polymerase (TaqNovaHS, Blirt, Poland), and 50 ng of genomic DNA as a template. A typical PCR procedure was as follows: 5 min at 95 °C, then 35 cycles of 30 s at 94 °C, 30 s at 50–60 °C (depending on the primer, Table 1), 1 min

at 72 °C, and 5 min at 72 °C. 0.5µL Midori Green Direct (Nippon Genetics Europe) was added to each amplification product, ran on 2% agarose gel (Sigma), and then visualized and photographed (EZ GelDoc System, BioRad).

**Table 1.** Primer sequences and PCR conditions used for *Lr54 + Lr37* marker identification on 2S$^k$ chromosome.

| Marker | Primer Sequence (5′ to 3′) | Amplification Temperature (°C) | Amplicon Size (bp) | Reference |
|---|---|---|---|---|
| S14-297 | CATGCAGAAAACGACACACC | 60 | 297 | [21] |
| | GGTAAGTGGTCAGGCGTTGT | | | |
| S14-410 | ACCAATTCAACTTGCCAAGAG | 61 | 410 | [22] |
| | GAGTAACATGCAGAAAACGACA | | | |

## 3. Results

### 3.1. Chromosome Segregation in 40 + M2R + M2S$^k$ Plants of Triticale

In the control experiment, pollen mother cells (PMCs) of ten triticale plants carrying 40 + M2R + M2S$^k$ chromosomes were examined to identify metaphase I (MI) and anaphase I (AI) of meiosis (Figure 2a). We observed a 2S$^k$ univalent in all of 100 analyzed PMCs (Figure 2b). The number of misdivided 2S$^k$ chromosomes during AI was 9 (Figure 2c). No other misdivided chromosome of triticale was observed. Next, 48 spikes of the monosomic substitution plants (40 + M2R + M2S$^k$) were allowed to self-pollinate, and 2601 seeds were obtained. A subset of 289 plants were screened using GISH and none were found with the expected RobT. 185 (63.1%) of the derived plants carried a whole *Aegilops* chromosome. In the remaining offspring (104 plants; 36.9%), no *Ae. kotschyi* chromosome fragments were identified.

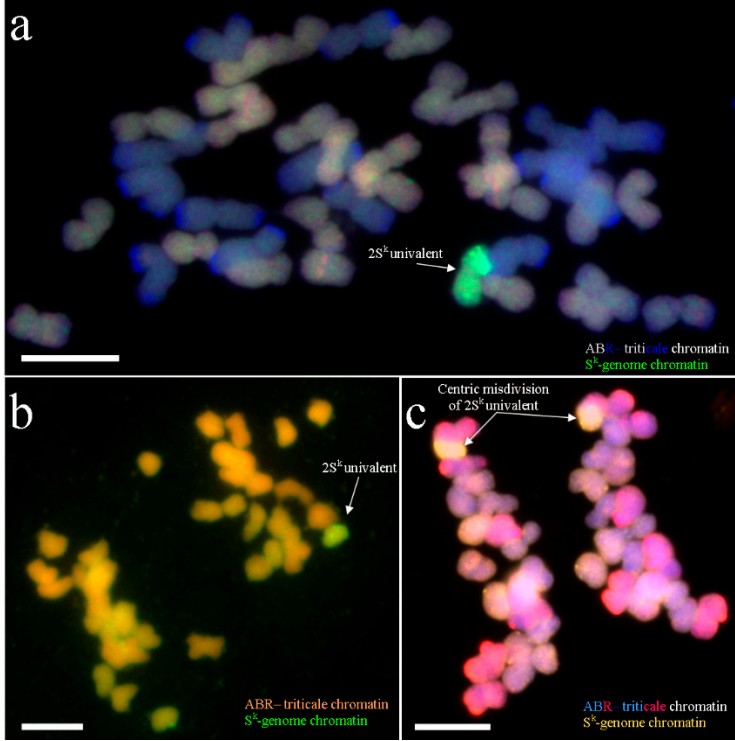

**Figure 2.** (**a**) Chromosome set of a 40 + M2R + M2S$^k$ triticale plant (**b**) Segregation of chromosome 2S$^k$ (green, Atto-488) at anaphase I in a 40 + M2R + M2S$^k$ triticale plant. (**c**) Centric misdivision of chromosome 2S$^k$ (yellow, Atto-647) at anaphase I in a 40 + M2R + M2S$^k$ triticale plant. Scale bar, 10 µm.

### 3.2. Chromosome Segregation in 40 + M2RS + M2RL + M2S$^k$ Plants of Triticale

The main goal of the experiment was to obtain triticale plants with introgression of a 2S$^k$ chromosome segment on the 2R chromosome. Hence, we crossed monosomic substitution (40 + M2R + M2S$^k$) plants with double ditelosomic D2RSD2RL plants of triticale to obtain triple monosomic triticale plants (40 + M2RS + M2RL + M2S$^k$). These plants contained a single telosomic 2RS, a single telosomic 2RL and a single monosomic 2S$^k$ chromosome (Figure 1b). 13,457 flowers of the ditelosomic line were pollinated with pollen collected from monosomic substitution M2S$^k$ (M2R) plants, and 3464 seeds were obtained (25.74% crossing efficiency). The chromosomal constitution of 100 randomly chosen plants were examined by a combination of genomic and fluorescence in situ hybridization methods (GISH/FISH), using total genomic DNA of rye and *Aegilops* species, and pTa-103 centromere sequence, as probes. 38 plants were found to be triple monosomic (40 + M2RS + M2RL + M2S$^k$) (Figure 3), 35 lacked the 2R chromosomes (40 + N2R + M2S$^k$), 15 plants were ditelosomic (40 + M2RS + M2RL) and 12 plants were double telosomic (40 + D2RL).

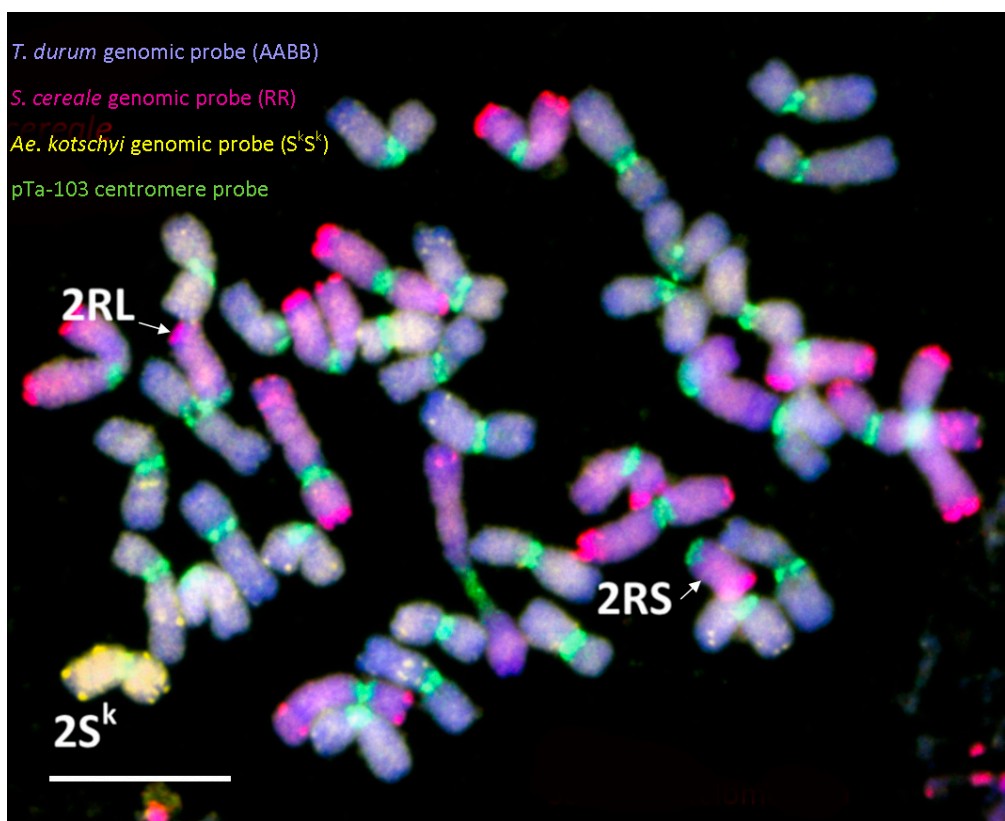

**Figure 3.** Karyotype of triticale plant carrying 40 + M2RS + M2RL + M2S$^k$ chromosomes, examined by genomic/fluorescence in situ hybridization (GISH/FISH). Total genomic DNA of rye (red) and *Ae. tauschii* (yellow), and pTa103 centromere sequence (green) were used as probes. Scale bar, 10 μm.

The triple monosomics were selected from further analysis (38 plants; 40 + M2RS + M2RL + M2S$^k$), and MI and AI were observed in 10 PMCs of each plant. Chromosome 2S$^k$ was present as a univalent in all PMCs examined. In most cases (35 plants; 92.11%), the *Aegilops* univalent underwent chromosome segregation and was moved randomly to one of the opposite poles of the cell (Figure 4a).

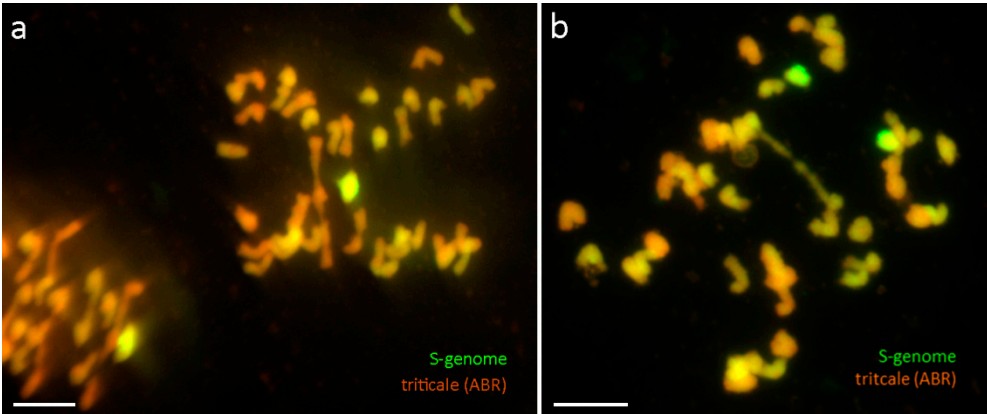

**Figure 4.** (**a**) Chromosome segregation of 2S$^k$ (green, Atto-488) at anaphase I of triticale line 40 + M2RS + M2RL + M2S$^k$. (**b**) Centric misdivision of chromosome 2S$^k$ (green, Atto-488) at anaphase I of triticale line 40 + M2RS + M2RL + M2S$^k$. Scale bar, 10 μm.

We also observed the misdivision of the 2S$^k$ chromosome arms in three plants (7.89%) (Figure 4b). Next, a total number of 68 spikes of 38 offspring plants (40 + M2RS + M2RL + M2S$^k$) were self-pollinated, and 3489 seeds were obtained. The chromosome constitutions of 100 randomly chosen plants were examined by FISH/GISH, using repetitive sequences as probes (pTa-86; pTa-535 and total genomic DNA from rye). The results of chromosome 2S$^k$ misdivision were observed in 13 plants, including: a double RobT-T2RS.2S$^k$L and T2S$^k$S.2RL (1 plant; Figure 5a, b and c); six plants carrying T2RS.2S$^k$L, four plants with T2S$^k$S.2RL, and two telosomic plants with the 2RS, 2RL and 2S$^k$S chromosome arms (Table 2).

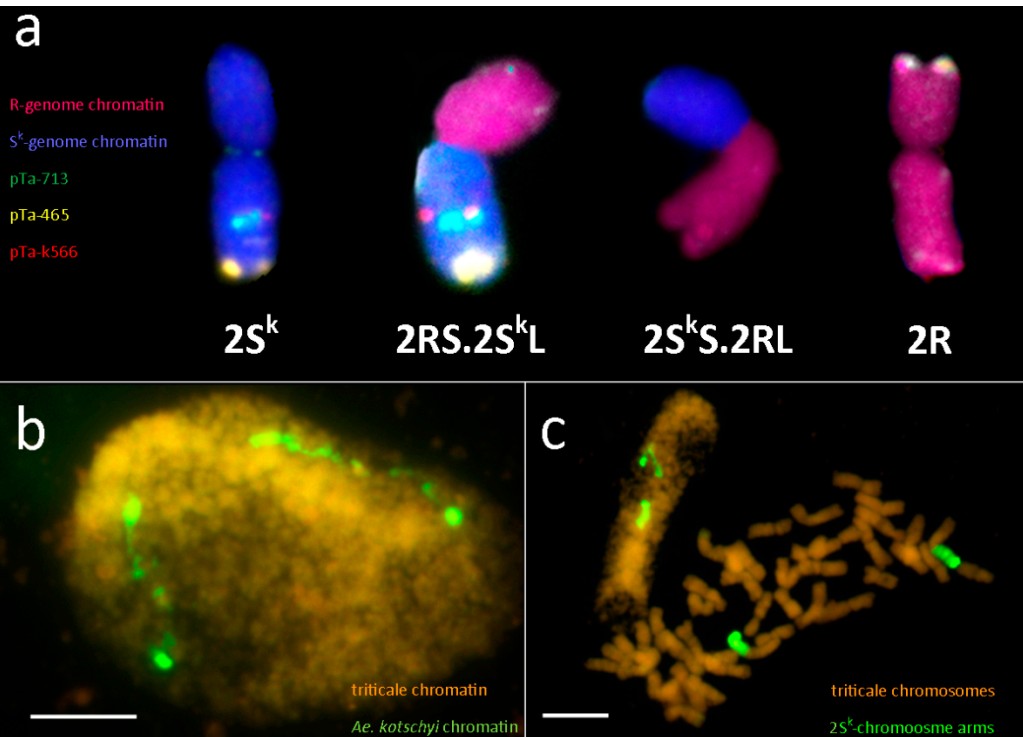

**Figure 5.** (**a**) T2S$^k$.2R Robertsonian translocations in triticale analysed by fluorescence in situ hybridization (R-genome chromatin—purple, S$^k$-genome chromatin—blue, pTa-713—green, pTa-465—yellow, pTa-k566—red). Genomic in-situ hybridization pattern of triticale plant carrying Robertsonian traslocations with segments of 2S$^k$ chromosome (green) at prophase (**b**) and metaphase (**c**) of root meristem cell. Scale bar, 10 μm.

**Table 2.** Chromosome constitutions and Lr54 + Yr37 SSR marker analysis of triticale plants carrying divided 2S$^k$ chromosome.

| Plant No. | Chromosome Constitution | Chromosome Number | Amplicon Size for S14-410 Marker (bp) | Amplicon Size for S14-297 Marker (bp) |
|---|---|---|---|---|
| 1 | 40 + M2RS + M2RL + M2S$^k$S | 43 | null | null |
| 2 | 40 + T2RS.2S$^k$L + M2RL | 42 | 410 | 297 |
| 3 | 40 + T2S$^k$S.2RL | 41 | null | null |
| 4 | 40 + T2RS.2S$^k$L | 41 | 410 | 297 |
| 5 | 40 + T2RS.2S$^k$L + M2RL | 42 | 410 | 297 |
| 6 | 40 + T2RS.2S$^k$L + t2S$^k$S.2RL | 42 | 410 | 297 |
| 7 | 40 + T2S$^k$S.2RL | 41 | null | null |
| 8 | 40 + T2RS.2S$^k$L | 41 | 410 | 297 |
| 9 | 40 + M2RS + M2RL + M2S$^k$S | 43 | null | null |
| 10 | 40 + T2RS.2S$^k$L | 41 | 410 | 297 |
| 11 | 40 + T2S$^k$S.2RL + M2RS | 42 | null | null |
| 12 | 40 + T2RS.2S$^k$L + M2RL | 42 | 410 | 297 |
| 13 | 40 + T2S$^k$S.2RL | 41 | null | null |

### 3.3. Lr54 + Yr37 SSR Markers Analysis

Initial analysis of molecular markers S14-297 and S14-410, linked to Lr54 + Yr37 loci [21,22], was conducted on the parental lines; namely the monosomic substitution M2S$^k$ (M2R) line and double ditelosomic D2RSD2RL triticale line. The same protocol was used to examine the subsequent offspring. An amplicon of 297bp (base pairs) was for obtained for the S14-297 marker, and 410bp for the S14-410 marker, as observed in control samples of *Ae. kotschyi* and double 40 + M2R + M2S$^k$ monosomic plants (Figure 6). The same 410bp product was amplified form the 6 plants with T2RS.2S$^k$L RobTs and for 1 plant carrying double RobTs-T2RS.2S$^k$L and T2S$^k$S.2RL (Figure 6; Table 2).

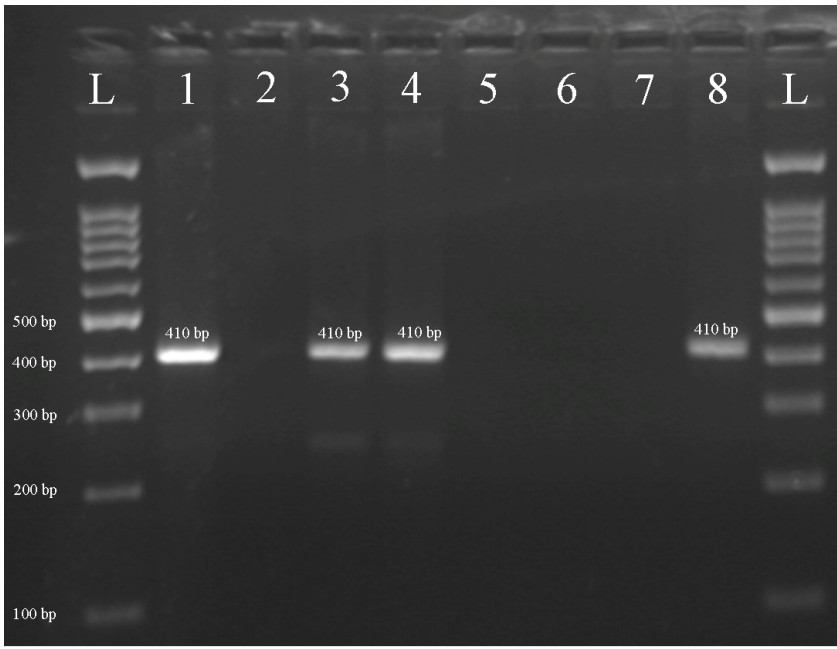

**Figure 6.** Amplification products of S14-410 marker linked to loci of *Lr54 + Lr37* leaf and stem rust resistance genes. (**1**) *Aegilops kotschyi* (UUS$^k$S$^k$); (**2**) triticale 'Sekundo'(AABBRR); (**3**) double 40T + M2R + M2S$^k$ monosomic; (**4**) T2RS.2S$^k$L and T2S$^k$S.2RL; (**5**) double ditelosomic D2RS + D2RL; (**6**) telosomic M2RS + M2RL + M2S$^k$S; (**7**) T2S$^k$S.2RL; (**8**) T2RS.2S$^k$L.

## 4. Discussion

Many reports have shown that RobTs in wheat can arise by centromeric misdivision of univalents at AI, followed by segregation of the derived telocentric chromosomes to the same nucleus, and fusion of the broken parts [26,30,31]. The focus of this study was to exploit the formation of RobTs to transfer of the *Lr54* + *Yr37* gene loci from *Ae. kotschyi* into triticale. We have focused on two essential aspects of this process: the frequency of univalent misdivision, and the frequency at which the broken chromosomes fuse to form a RobT. The control experiment used the standard approach of self-pollinating monosomic substitution (40 + M2R + M2S$^k$) of triticale. Genomic in-situ hybridization identified that chromosome 2S$^k$ was present in all meiotic cells assayed, and the centromere was divided during AI in 9% of the PMCs surveyed. The same event was observed in PMCs isolated from the 40 + M2RS + M2RL + M2S$^k$ plants. The frequency of PMCs where the centromere of the 2S$^k$ univalent divided was 7.89%. Our results are comparable with other reports, showing that the frequency of univalent misdivision range between 3% up to 51% [26,30,31]. When both approaches were compared (40 + M2R + M2S$^k$ vs 40 + M2RS + M2RL + M2S$^k$), no significant difference in the frequency at which the *Aegilops* univalent misdivided was observed. However, the 2R univalent did not misdivide in the control experiment, and the use of the double ditelosomic D2RS + D2RL in the alternative approach was justified.

It can be assumed that the probability of RobTs formation is dependent on the rate of centromere breakage, and subsequent formation of telocentric chromosomes, occurs during meiosis. In wheat, the frequency of recovery of wheat–alien RobTs is between 2% and 20%, depending on the chromosomes involved [23,26,30,31,40–44]. In our experiment, the progeny of monosomic substitution plants lacked any kind of triticale-*Aegilops* translocation. This could correlate with the lack of the 2R chromosome misdivision observed in these plants. Similar issues were reported by Ghazali et al. [45], who aimed to produce whole arm RobTs involving chromosomes 2B (wheat) and 2E$^b$ (*Thinopyrum bessarabicum*). Surprisingly, the authors obtained only T2E$^b$S.2BL, while no 2BS.2E$^b$L RobTs were recovered [45]. It has been reported that rye chromosomes rearrange at a higher rate when compared to wheat chromosomes [46], with the long arm of chromosome 2R more readily fused with the chromosome arm of an alien species. For example, Fiebe et al. [47] derived 2BL.2RL RobTs carrying Hessian fly resistance gene *H21* in wheat cv. Hamlet. Rahmatov et al. [48] also obtained 2DS.2RL RobT, which transfered the stem rust resistance gene *Sr59* into wheat.

In our study, we have effectively used ditelosomic lines of triticale to increase the frequency of 2S$^k$.2R RobTs generated. Moreover, we have obtained both 2S$^k$S.2RL and 2RS.2S$^k$L RobTs. The ability to produce both types of RobTs was an important goal as the particular chromosome segments of *Aegilops* may harbor other desirable genes. To summarize, we obtained plants with chromosome translocations, which probably carry the *Lr54* + *Lr37* leaf and stem rust resistance genes. The next step of this project will be to phenotype the eleven RobTs lines for leaf rust or stripe rust resistance. If these tests succeed, such germplasm could be a valuable source for triticale resistance breeding.

**Author Contributions:** Conceptualization, M.T.K.; methodology, M.T.K.; validation, W.U.; formal analysis, W.U.; investigation, W.U., J.B., J.S. and R.S.; resources, M.T.K., H.W.; data curation, M.T.K., W.U.; writing—original draft preparation, W.U., M.T.K.; writing—review and editing, W.U., D.P. and M.T.K.; visualization, W.U., M.T.K.; supervision, M.T.K., D.P. and H.W.; project administration, M.T.K.; funding acquisition, M.T.K.

**Funding:** This research and the APC were funded by NATIONAL CENTRE FOR RESEARCH AND DEVELOPMENT, Poland (Narodowe Centrum Badań i Rozwoju, Polska), grant number LIDER/3/0004/L-8/16/NCBR/2017.

**Acknowledgments:** We would like to acknowledge the administrative support provided by the Institute of Plan Genetics of the Polish Academy of Sciences.

**Conflicts of Interest:** The authors declare no conflict of interest.

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
