# Peer review of "Recovery of 2R.2Sk Triticale-Aegilops kotschyi Robertsonian Chromosome Translocations"

_agronomy, doi:10.3390/agronomy9100646_

Round 1

Reviewer 1 Report

Authors have done a good job to correct their paper. Reviewer's suggestions and recommendations have been well considered and taken as basis to improve the manuscript.

Author Response

Many thanks for your review and all comments and clues.

We have corrected all of minor spelling mistakes and edits.

Kind regards,

Michał Kwiatek

on behalf of co-authors

Reviewer 2 Report

The authors clearly show that use of double ditelosomic chromsomes is helpful in generating Robertsonian translocations from an Ae. kotschyi group 2 chromosome.  I recommend removing the reference to rust resistance in the title since this is not a part of this study. I only have minor suggested edits in the attached revised document.

Author Response

We would like to thank for your advices, corrections and clues.

As recommended, we have removed the reference to rust resistance in the title.

We have made all of suggested edits, which were pointed in attached revised document.

Kind regards,

Michał Kwiatek

on behalf of co-authors

This manuscript is a resubmission of an earlier submission. The following is a list of the peer review reports and author responses from that submission.

Round 1

Reviewer 1 Report

This paper fully fits to the subject area no. 1 "Crop genetics and breeding" of this journal.

The paper aims at the systematic development of translocation(s) of triticale using Ae. kotschyi resistant source to establish "new" leaf and stripe rust resistance in triticale.

This work comprises sophisticated cytogenetic analyses of the experimental materials, e.g. including recombined Triticum-Aegilops chromosomes.

The presence of Lr54+Lr37 resistance loci is indicated by respective marker analysis. However, the effectiveness of these loci with regard to leaf and stripe rust resistance has not been confirmed or described in the present paper (see title and goal). 

This fact needs to be indicated in the paper title (e.g. "...potentially conferring rust resistance in triticale").

This interesting work focusing on cytogenetic analyses of potentially useful basic breeding material may deserve publication subject to major corrections and improvement of the manuscript (see attached comments).

Author Response

Many thanks for your review, which provides valuable comments and clues.

This communication article shows the ways of generation of Robertsonian translocations. Hence, we shortened and changed the title by adding “potentally”. At this stage we have only several  plants carrying RobTs. The small number of plants does not allow to conduct phenotype analysis (inoculation tests). We are going to reproduce these plants and carry out multiple replication experiments on a large number of plants. This will be done in the next step of the project.

 Respond for detailed comments:

Title was changed as mentioned above.

The paper of Sodkiewicz et al. was cited to show, that our research group is involved in development of genetic variability of triticale considering leaf rust resistance. However, in accordance to your suggestion, we have eliminated this citation from the MS.

There is no data showing how the 2D.2Sk wheat-Ae. kotschyi translocation lines was used in wheat breeding and there is no other sources indicating the use of Lr54+Yr37 molecular markers in wheat or triticale breeding.

We have deteled Sodkiewicz et al. 2004.

We have deleted the sentence: “All hybrid plants were obtained and assembled at the IPG PAS.”

We have added a year of cross-hybridization trials (2018).

We have corrected the results showing the number of plants with RobTs. In the initial version our MS the percentage was mistaken for the number of plants. We have rearranged this section and now we hope, that this will be clear for the readers. Moreover, we added Table 2 to present detailed results.

We have corrected the typos in “Robertsonian translocation” phrase.

The text was verified by our US collaborator.

Kind regards,

Michał Kwiatek

Reviewer 2 Report

This manuscript documents alternative approaches to producing Robertsonian translocations of the Ae. kotschyi 2S into triticale.  

The clarity of the manuscript would be improved by an ideogram (figure) outlining the two contrasting approaches.

Two questions arose during review:

Beginning at L173:  38 triple monosomics were used.  And 10 pollen mother cells from each plant were analyzed.  But somehow these 38 triple monosomics became 92 plants (L176) plus 8 plants (L184), which mysteriously adds up to 100 plants.  Clarification is required here. It appears (L188, L209) that the authors have recovered a total of 11 RobT's.  It appears that all 11 RobTs amplified the target product for primers associated with Lr54+Yr37, whether they were 2SkS or 2SkL RobTs.  This is perplexing because the gene complex should be on one arm or the other, not both.  There needs to be some explanation provided for this observation.  Either the markers are not working, or the discrimination of the two classes of RobTs is incorrect.

Finally, the manuscript would have been considerably strengthened by phenotyping the 11 RobTs for either leaf rust or stripe rust or both.  Given the perplexing result of the PCR analysis, and the lack of phenotypic data, the title of the manuscript should be revised to remove reference to the two rust resistance genes.

Author Response

At first, we are grateful for comments and clues. All of them play a positive role in improving our article.

As you’ve suggested, we have added Fig.1 which shows the two pathways of RobTs induction.

We have also corrected the number of plants. In the initial version our MS the percentage was mistaken for the number of plants. We have rearranged this section and now we hope, that this will be clear for the readers. Moreover, we added Table 2 to present detailed results.

We have repeated the PCR analyses of plants with different variants of chromosome aberrations. We have changed the figure 4 which is now more informative for the readers.

In the discussion we have pointed the advantages of the use of double ditelosomic line of triticale in the context of  chromosome arm fusions, which result in the centric translocations production, that have the breakpoint at the centromere.

We have obtained a very few plants carrying desirable chromosome translocation, hence, at this step, we were not able to proceed the phenotypic analyses. We are aware that this is the initial study, so we changed the title. The next step for this experiment is to obtain the numerous offspring and conduct the inoculation tests.

The text was verified by our US collaborator.

Kind regards,

Michał Kwiatek

Round 2

Reviewer 2 Report

The authors have addressed the principal concerns that I had in the original review.  The addition of Figure 1 makes the manuscript more approachable.

The manuscript would be strengthened by phenotyping for rust resistance.